# Randomized Trial of Feasibility and Preliminary Effectiveness of PerioTabs® on Periodontal Diseases

**Begum Alkan** [1,2,*] and **Mutlu Özcan** [3]

1   Private Practice of Periodontology, Istanbul 34353, Turkey
2   Department of Periodontology, Faculty of Dentistry, Istanbul Medipol University, Istanbul 34250, Turkey
3   Center or Dental Medicine, Division of Dental Biomaterials, Clinic for Reconstructive Dentistry, University of Zurich, 8032 Zurich, Switzerland; mutluozcan@hotmail.com
*   Correspondence: alkan.bgm@gmail.com

**Abstract:** This double-blinded and split-mouth design, randomized feasibility study aimed to assess the efficacy of prophylactic and therapeutic use of a new age NitrAdine™-based brushing solution (PerioTabs®) on the clinical parameters before and after periodontal therapy. Four subjects were randomly assigned to two treatment groups according to periodontal infection: PerioTabs® and placebo. At the first appointment, a split-mouth scaling was performed in all participants. All oral care instructions concerning the use of effervescent tablets were explained to the participants in detail. The morning after the tablets were finished, a full-mouth scaling was completed in all participants. All parameters were recorded at 0, 11, and 40 days. According to the results, both the therapeutic and prophylactic interventions showed similar impacts on the gingival index and probing depth compared to the placebo in all patients. The bleeding on probing was further reduced at the PerioTabs® group at baseline until day 40 compared to the placebo in the gingivitis patients with both interventions; at both 11 and 40 days in the periodontitis patients receiving the therapeutic intervention; and at baseline to day 40 in the periodontitis group receiving the prophylactic intervention. While the preliminary results of this new-age brushing solution appear to be a promising approach for a future therapy of periodontal diseases, further research on a larger sample size is needed to draw firm conclusions.

**Keywords:** adjunctive treatment; antimicrobials; non-invasive treatment; periodontology; scaling; root planing

## 1. Introduction

Periodontal inflammation is a significant health problem due to its high prevalence worldwide [1]. Periodontal health means more than the quality oral health; it is also an integral part of general health. Dental plaque-induced gingivitis caused by inadequate oral hygiene is a reversible inflammatory reaction, and the primary clinical parameter of gingivitis is excepted bleeding upon gentle probing in the last classification of periodontal conditions [2]. If left untreated, pathogenic microorganisms may persist in the infected tissue and contribute to loss of periodontal attachment and alveolar bone, and so in most instances, gingivitis is accepted as the first stage in the development of periodontitis [3,4]. The prevention of periodontitis involves both the treatment of existing gingivitis and the prevention of future occurrences.

Initial periodontal therapy consists of non-surgical periodontal therapy, including supragingival calculus removal, tooth surface polishing, scaling and root planing (SRP), the elimination of plaque retentive factors, and oral hygiene advice to facilitate the long-term maintenance of healthy periodontal tissue. SRP is the most commonly used treatment approach in initial periodontal therapy, but clinical treatment alone is futile if the patient is unable to implement adequate oral hygiene. The oral cavity is the beginning of the gastrointestinal tract and is continuously subjected to internal and external influences,

revealing various physical, chemical, microbiological, and thermal stimuli. The composition of the oral biofilm that covers the entire mouth also varies from region to region and within a region example on the teeth, tongue, and gingiva. The periodontal ligament is a unique tissue that connects the cementum to the alveolar bone, and its inflammation is not limited to the oral cavity but also negatively affects general health through systemic circulation. All of these factors and their interactions demonstrate the importance of successful SRP. The development of new products that support the treatment of oral diseases continues to be one of the most popular research areas in periodontology today.

PerioTabs® is a new-generation antimicrobial gingival brushing solution that is supplied in the form of effervescent tablets (one tablet per day for ten days) that was developed as an adjunct to initial periodontal treatment and can be self-administered by the participants at home during the research period. The tablets have antibacterial, antifungal, and antiviral properties. Several in vitro studies have demonstrated the antimicrobial effects of NitrAdine™ on *Candida species* [5–7], *Staphylococcus aureus* [5–8], and *Escherichia coli* [7] as well as its antiviral effects on *Herpes Simplex Virus 1* [8]. The positive effects of an antimicrobial emulsion containing NitrAdine™ for the disinfection of oral medical appliances have been demonstrated [5,6,9–12]. Moreover, a new-generation NitrAdine™-based periodontal dressing combined with a gum brushing solution has recently been introduced to reduce the presence of microorganisms, such as *Aggregatibacter actinomycetemcomitans*, *Porphyromonas gingivalis*, and *Prevotella intermedia*, in the oral cavity, promoting wound healing following periodontal therapy [13,14]. These studies have yielded promising results. Although randomized clinical trials of NitrAdine™-based materials have been conducted, no randomized clinical trials of participant management following SRP alone have been published to date.

The dependence of oral hygiene practices on study participants complicates the accurate interpretation of clinical research on oral hygiene products. Therefore, feasibility studies are essential to identify trial weaknesses before randomized clinical design studies are conducted. This feasibility trial investigated participant compliance with the interventions to provide data for the purpose of estimating the parameters required in designing a trial. It also aimed to determine whether a randomized clinical trial of the prophylactic brushing solution as an adjunct treatment before and after SRP constituted an appropriate trial design.

The primary objectives of this randomized, blinded, placebo-controlled, split-mouth trial were as follows: (1) to assess the efficacy of the exclusive use of the solution on clinic parameters; (2) to evaluate the efficacy of the prophylactic use of the solution on clinical parameters after SRP; (3) to determine whether the solution can enhance the therapeutic effects of SRP in patients with periodontal disease; (4) to compare the efficacy of the prophylactic and therapeutic use of the solution on the clinical parameters after SRP; and (5) to evaluate possible side effects, such as allergic reactions, gingival irritation, gingival pain, gingival bleeding, halitosis, xerostomia, and satisfaction. The study's secondary objectives were (1) to observe whether a split-mouth design is an appropriate methodology for a brushing solution; (2) to pilot PerioTabs® effervescent tablets; (3) to pilot placebo effervescent tablets; and (4) to observe the material–method requirements for a definitive trial, such as the time required to reach the target data, the acceptance rate of the eligible participants invited to take part, whether the eligibility criteria set for the participants are too open or too restrictive, the participants' willingness to attend appointments, and whether they understand the oral hygiene instructions described.

To the best of our knowledge, no randomized clinical trial to date has evaluated the effects of the PerioTabs® on periodontal healing in participants with different periodontal conditions. The aim of this trial was to observe and compare the efficacy of PerioTabs® on clinical parameters as an adjunctive agent before and after periodontal therapy and to predict the material–method requirements for a definitive clinical trial.

## 2. Materials and Methods

We conducted a prospective, interventional, single-center, double-blinded, parallel-armed, placebo-controlled, and split-mouth designed, randomized feasibility clinical trial. The study protocol was conducted according to the guidelines of the Declaration of Helsinki of 1975, revised in 2013, and approved by both the Istanbul Medipol University Clinical Research Ethics Committee (Number: E-66291034-604.01.01-2798, Date: 15 June 2021) and the Turkish Medicine and Medical Devices Agency (Number: E-68869993-511.06-491724, Date: 28 July 2021). Written informed consent, including the study protocol and information that the study would be published in an international journal, was obtained from all participants at the beginning of the study.

A total of four eligible participants diagnosed according to the new classification framework proposed by the 2017 World Workshop on the Classification of Periodontal and Peri-Implant Diseases and Conditions [15–17] were recruited from the Faculty of Dentistry, Department of Periodontology in Istanbul, Turkey, between August and October 2021. Participants who satisfied the following criteria were included in the trial: age between 18 and 65 years; willingness to use only oral care products provided by the investigators during the trial; adverse history of any infectious or systemic disease; negative history of allergy to persulfates; non-smokers and non-drinkers; having more than 20 teeth; and no previous treatment for periodontitis and gingivitis. Participants with the following characteristics were excluded from the trial: age under 18 or over 65 years; pregnant or lactating females; history of systemic or infectious disease; history of allergy to NitrAdine$^{TM}$; smoking or use of tobacco in any form; alcoholism; the use of antibiotics and anti-inflammatory drugs within six months prior to the start of the trial; fewer than 20 teeth; the use of orthodontic appliances or removable prosthetic appliances; and the use of daily chemical plaque inhibitors or mouthwash. In August 2021, all of the patients referred to the clinic were screened, and potential volunteers who met the inclusion criteria were identified. The written informed consent form, which included information about the purpose of the trial, the procedures to be carried out, and the participant's rights, was signed by all participants prior to their enrollment in the trial.

This trial was designed for participants suffering from different periodontal diseases. Two participants diagnosed with gingivitis on an intact periodontium associated with dental biofilm (G) and two participants diagnosed with stage III grade B generalized periodontitis (P) were recruited for the trial. The participants received either NitrAdine$^{TM}$ (PerioTabs$^{®}$, Bonyf AG, Vaduz, Liechtenstein) or placebo, and half the mouth was randomly exposed to therapeutic (NitrAdine$^{TM}$/placebo treatment after SRP) or prophylactic intervention (NitrAdine$^{TM}$/placebo treatment before SRP). At the first appointment, intraoral photos were made, clinical periodontal parameters including plaque index (PI) [18], gingival index (GI) [19], probing depth (PD), bleeding on probing (BOP), gingival recession, and clinical attachment level (CAL) were recorded from six sites (disto-buccal, mid-buccal, mesio-buccal, disto-palatal, mid-palatal, and mesio-palatal) around each tooth using William's periodontal probe (Hu-Friedy, Chicago, IL, USA) immediately before SRP. Oral hygiene instruction was demonstrated via a modified Stillman's brushing technique and an interdental brush using a cast model. SRP was performed on half of the mouth randomly, which had been divided into the right and left quadrants. After SRP, randomized NitrAdine$^{TM}$ or placebo effervescent tablets were used to brush the teeth and gums for 2 min, once a day, for 10 days. Intraoral photos and clinical periodontal parameters were recorded at baseline (day 0), immediately prior to any treatment, on day 11, subsequent to SRP and effervescent tablets application, and on day 40, the end of the trial. All participants received a kit (PerioTabs$^{®}$ or placebo) containing ten small effervescent tablets (one tablet per day for ten days). One tablet was prepared by dissolution in 15 mL of lukewarm water in a pre-calibrated container, provided along with the kit, to create a brushing solution. To use the solution, a new toothbrush should be immersed in the solution for 15 min to allow the tablet to dissolve completely and for the toothbrush bristles to absorb the solution. Once the tablet is completely dissolved, the teeth and gums are brushed gently using a

toothbrush immersed in the solution. Throughout the procedure, it is recommended that the toothbrush be immersed in the solution 2–3 times for a few seconds. Participants should brush every evening after eating. After brushing, participants should rinse their mouths thoroughly with water. Participants were advised not to use any other toothpaste while using the solution. All instructions for use were explained in detail, and a user guide manual was given to the participants. All measurements were conducted by the same blinded investigator throughout the trial.

PD reduction was the primary outcome variable for the periodontitis patients, while the reduction of gingival inflammation parameters were the primary outcome variables for the gingivitis patients.

Since this was a feasibility trial, no sample size calculation was performed. The researchers aimed to complete the trial with four participants because this was considered a reasonable sample size to obtain a preliminary perspective on trial design and on how long each participant who met the inclusion criteria per group would be available.

Participants were numbered according to their time of application to the clinic and were randomly allocated to one of the two effervescent tablet groups using a referee flipping a coin. The participants' identities were kept confidential. The participants and the researchers performing the treatment, collecting data, and evaluating the outcomes were blinded to allocation.

Changes in clinical periodontal parameters at each site were investigated at the baseline, on day 11, and on day 40. All data were stored in a personal computer using Excel software and checked to exclude false values and identify missing values. GI scores were evaluated following Smith et al. [20]. The percentage of sites with PD $\geq$ 5 mm was calculated. CAL was calculated by adding the measurement for PD to that of gingival recession at each site, and the CAL percentage was calculated similarly to the PD. To compare the changes in clinical periodontal parameters among different treatment groups of the same disease, the table data were converted to 3D column charts, yielding enhanced visualization. Percentage results were used to interpret the differences in the clinical periodontal parameters at each time point within and between the treatment groups.

## 3. Results

A total of four subjects (G, n = 2; P, n = 2) who met the eligibility criteria were randomly assigned to the PerioTabs® or placebo group, having agreed to receive the intended split-mouth design treatment (therapeutic or prophylactic intervention), and they were evaluated. All of the participants attended all of the appointments and completed the study. Participant enrollment commenced and was completed in August 2021. Data collection was conducted between August 2021 and October 2021. Participants were evaluated at the baseline and follow-up days 11 and 40. This feasibility trial ended when the target number of participants completed the study and sufficient data had been obtained.

Table 1 shows the baseline demographic and clinical characteristics for each group. The distribution of the diagnostic data is similar between the groups according to medication, and no differences were observed in the clinical periodontal parameters between the participants in their corresponding groups at baseline (Figure 1). An investigator examined each patient who attended the clinic to identify the eligible participants within a four-week period. All four eligible participants who attended our clinic agreed to participate in the study.

Figure 2 shows the results of both the therapeutic and prophylactic interventions using PerioTabs® compared to placebo in G patients. The efficacy of the (1) PerioTabs® alone on the clinic parameters in percentage (%) appears to be similar to that of placebo (PI: 64 vs. 79, 25 vs. 18; GI: 27 vs. 27, 4 vs. 2; PD: 7 vs. 0, 4 vs. 0; BOP: 70 vs. 68, 52 vs. 34; CAL: 7 vs. 0, 4 vs. 0; respectively); (2) prophylactic use of PerioTabs® on PI, PD, BOP, and CAL appears to be more successful than that of the placebo after SRP (PI: 64 vs. 79, 2 vs. 41; GI: 27 vs. 27, 0 vs. 4; PD: 7 vs. 0, 0 vs. 0; BOP: 70 vs. 68, 11 vs. 25; CAL: 7 vs. 0, 0 vs. 0; respectively); (3) therapeutic use of PerioTabs® as an adjunct to SRP on the BOP parameters appears to

be more successful than that of a placebo (PI: 50 vs. 91,11 vs. 45; GI: 29 vs. 54, 2 vs. 4; PD: 5 vs. 2, 0 vs. 0; BOP: 64 vs. 77, 11 vs. 32; CAL: 7 vs. 2, 0 vs. 0; respectively); and (4) the prophylactic and therapeutic use of PerioTabs® on the healing of the clinical parameters after SRP appears to show similar results (PI: 64 vs. 50, 2 vs. 11; GI: 27 vs. 29, 0 vs. 2; PD: 7 vs. 5, 0 vs. 0; BOP: 69 vs. 64, 11 vs. 10; CAL: 7 vs. 7, 0 vs. 0; respectively).

**Table 1.** Baseline demographic and clinical characteristics.

|  | NitrAdine™ (n = 2) | Placebo (n = 2) |
|---|---|---|
| Mean age (years) | 32.5 | 26 |
| Sex | | |
| Female | 2 (100%) | 1 (50%) |
| Male | 0 (0%) | 1 (50%) |
| Periodontal status | | |
| Gingivitis | 1 (50%) | 1 (50%) |
| Periodontitis | 1 (50%) | 1 (50%) |
| Smoking | 0 (100%) | 0 (100%) |
| Antibiotic consumption within 1 year | 0 (100%) | 0 (100%) |
| Periodontal therapy within 1 year | 0 (100%) | 0 (100%) |
| Systemic disease | 0 (100%) | 0 (100%) |

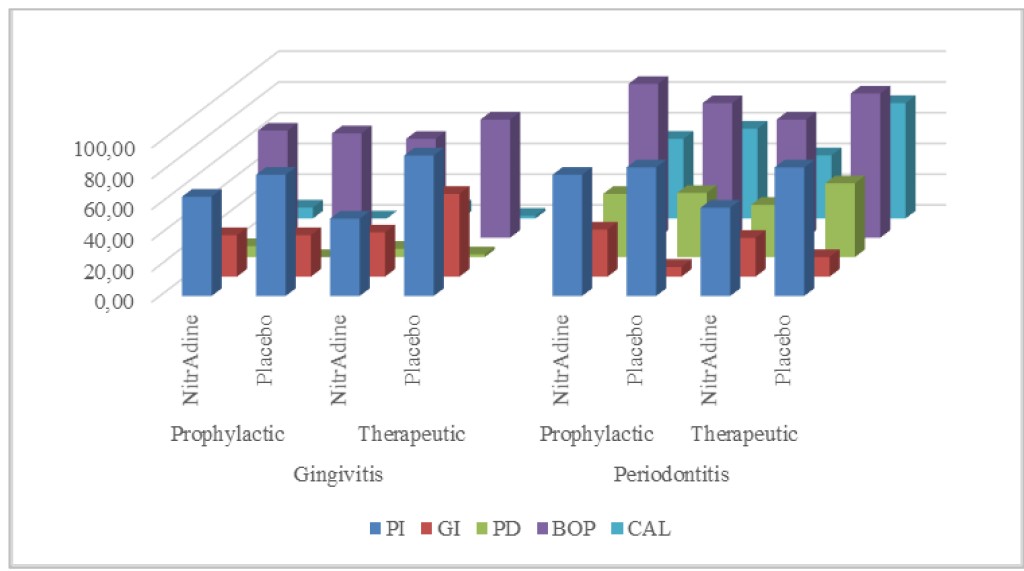

**Figure 1.** Baseline clinical periodontal parameters.

Figure 3 shows the results of both therapeutic and prophylactic interventions using PerioTabs® compared to the use of the placebo in the P patients. The efficacy of the (1) PerioTabs® alone on the clinic parameters in percentage (%) appears to be similar to that of the placebo. (PI: 79 vs. 83, 75 vs. 63; GI: 30 vs. 6, 25 vs. 6; PD: 41 vs. 42, 34 vs. 42; BOP: 100 vs. 88, 95 vs. 44; CAL: 52 vs. 59, 40 vs. 54; respectively); (2) the prophylactic use of the PerioTabs® on GI, BOP, and CAL appears to be more successful than the use of the placebo after SRP (PI: 79 vs. 83, 7 vs. 6; GI: 30 vs. 6, 2 vs. 6; PD: 41 vs. 42, 7 vs. 6; BOP: 100 vs. 88, 9 vs. 38; CAL: 52 vs. 59, 7 vs. 23; respectively); (3) the therapeutic use of the PerioTabs® as an adjunct to SRP on the BOP and CAL parameters appears to be more successful than that of the placebo (PI: 57 vs. 83, 4 vs. 2; GI: 25 vs. 13, 0 vs. 6; PD: 34 vs. 48, 2 vs. 4; BOP: 77 vs. 94, 0 vs. 35; CAL: 41 vs. 75, 2 vs. 33; respectively); and (4) the prophylactic and therapeutic use of the PerioTabs® on the healing of the clinical parameters after SRP

appears show similar results (PI: 79 vs. 57, 7 vs. 4; GI: 30 vs. 25, 2 vs. 0; PD: 41 vs. 34, 7 vs. 2; BOP: 100 vs. 76, 9 vs. 0; CAL: 51 vs. 41, 7 vs. 2; respectively).

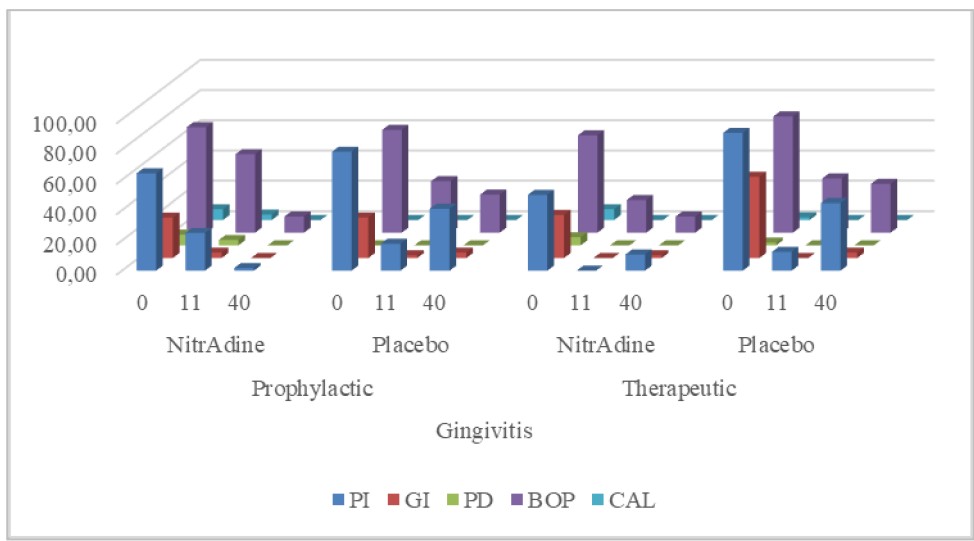

**Figure 2.** Alteration of periodontal parameters over time and among the G groups.

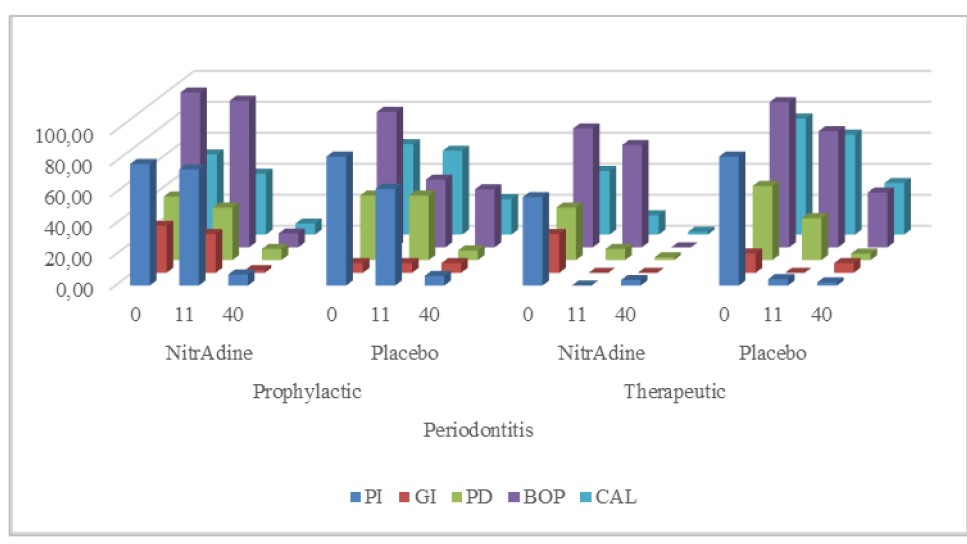

**Figure 3.** Alteration of periodontal parameters over time and among the P groups.

The PerioTabs® and placebo groups were similar with respect to age, gender, and systemic status, and no potential adverse or unintended effects were reported during the study period.

## 4. Discussion

The SRP technique used in the mechanical treatment of periodontal diseases can cause injury to the periodontal tissue. In addition to periodontal treatment, adjuvant agents can be used to reduce post-operative bleeding and pain, increase post-operative comfort, prevent bacterial colonization, and support wound healing. In the present study, we describe the use of a new gum brushing solution that is based on the antimicrobial NitrAdine™ formula, called "PerioTabs®". This split-mouth study was designed to determine the prophylactic and therapeutic efficacy of the periodontal brushing solution as an adjunctive agent to SRP on clinical periodontal parameters in patients with different periodontal diseases compared to the placebo brushing solution.

To increase the overall implications of the results, we included variability at the initial time point of use of the brushing solution and determined the number of follow-up days by considering both epithelial and connective tissue recovery time. Our data reflect the outcomes of this feasibility trial only; however, we hope that this study design may serve as a template for analyzing definitive clinical trials to understand the prophylactic and therapeutic effects of different oral agents. However, the limited number of participants and over-stringent exclusion criteria may limit the generalizability of the trial results. Moreover, the complexity of the study methodology may limit the clinical use and generalizability of this strategy for beginner researchers. Some investigators may struggle to provide the correct directions to the statistician and to understand and interpret which outcome data correspond to which questions.

One of the goals of this feasibility trial was to investigate the efficacy of PerioTabs® on clinical periodontal parameters as an adjunctive agent at different time points during periodontal therapy. The main results of the therapeutic and prophylactic intervention of the PerioTabs® compared to the placebo showed that the efficacy of the (1) individual use of PerioTabs® on the clinical parameters appears to be similar to that of the placebo; (2) prophylactic use of PerioTabs® on BOP and CAL appears to be more successful than the placebo after SRP; (3) the therapeutic use of the PerioTabs® as an adjunct to SRP on BOP parameters appears to be more successful than the use of the placebo; and (4) the prophylactic and therapeutic use of PerioTabs® on the healing of clinical parameters after SRP appears to show similar results for both G and P patients. These results showed that PerioTabs® effervescent tablets are effective in improving gingival and periodontal inflammatory parameters in both prophylactic or therapeutic use. However, we observed that in some cases, the PerioTabs® and placebo tablets showed similar effects. Therefore, we determined that a more ineffective placebo effervescent tablet should be prepared for definitive clinical trial.

To the best of our knowledge, our participants were able to accurately follow the oral hygiene instructions provided. Our results appear similar to those of earlier studies demonstrating the efficacy of NitrAdine™-based materials on clinical periodontal parameters. Sakly et al. [13] reported that NitrAdine™-based periodontal dressing and antimicrobial solution applied after SRP treatment showed significantly better healing on PI and gingival inflammatory parameters compared to the placebo group on days 5 and 11. Perelli et al. [21] declared that the adjunctive use of a PerioTabs® treatment in a daily oral hygiene routine appeared to be successful in reducing full-mouth bleeding upon probing score. Goguta et al. [22] demonstrated that PerioTabs® significantly reduced gingival inflammation in prosthetic patients compared to the control group. Cosola et al.'s [23] study, which involved patients with peri-implant mucositis, highlighted that both the Chlorhexidine and NitrAdine™ groups showed improvements in clinical periodontal parameters (excluding PD) after SRP and even showed that the GI and full-mouth bleeding scores for the NitrAdine™ group were significantly better than those of the Chlorhexidine group. On the other hand, Ashwini and Swatika [14] observed that both an SRP-plus-NitrAdine™-based periodontal dressing-plus-antimicrobial solution group and an SRP-only group showed statistically significant reductions in the GI, PD, and CAL values from baseline to follow-up visits at 30 days, but no significant differences were observed in the inter-group comparison at 30 days. Monje et al. [24] reported that the application of a periodontal dressing immediately after non-surgical mechanical therapy may be helpful in improving short-term clinical outcomes, but more controlled studies are needed to confirm this finding. The difference between the results of these two studies indicates that the follow-up times for monitoring periodontal wound healing, possibly after SRP, vary.

The participants were monitored for possible side effects from the beginning of the study until the last day of treatment. For spontaneously reported and directly observed side effects, anamnesis was taken from all patients at each clinical visit, and clinical examinations were performed. No patients stopped working due to side effects. The research process was well tolerated by all participants. In other clinical studies using NitrAdine™-based

materials [10,13,14], the participants reported no serious side effects. On the other hand, when evaluating the side effects, participants were asked whether they have any complaints about the solution during this process at each visit. We believe that a multiple-choice questionnaire would be more effective in assessing the participants' complaints regarding side effects.

People with a history of periodontitis are, by nature, at high risk for recurrence. Patients should be aware of this so that they can take necessary precautions and be more regular with their dental check-ups. In addition to the study treatment, periodontal therapy was performed by a professional dentist, and patients should achieve correct domiciliary oral hygiene techniques to maintain periodontal health status. In our study, the modified Stillman's brushing technique using a manual toothbrush and the correct use of interdental brush was shown to the patients. There have been studies reporting that optimal daily home care is achieved using toothpastes [25] containing both hyaluronic acid and lactoferrin and powered toothbrushes [26] with a rotating-oscillating head or sonic head. Patients could be re-informed about the importance of using the right toothpaste and an electrical toothbrush. The results of this feasibility study support the acceptability of a large randomized clinical trial involving patients with gingivitis or periodontitis for improving the results of SRP therapy. This result could be achieved even if all of the predetermined improvements are not always fully met. The achievement of predetermined results does not necessarily indicate the suitability of the research but rather highlights the required methodological changes. In some cases, the contributions of the PerioTabs® and results of the placebo effervescent tablets in periodontal recovery were similar. We believed that for gingival inflammatory parameters and PD, the NitrAdine$^{TM}$ formulation, which is known for its antimicrobial ability, would show significant effectiveness compared to the placebo.

A further aim of this trial was to observe the suitability of a split-mouth design methodology to test a brushing solution and the other material–method requirements for a definitive clinical trial. From a general perspective, the split-mouth design methodology of procedures to observe the results of different interventions appears to be successful. Conducting this feasibility trial in split-mouth design demonstrated that the results of six different placebo-controlled trials could be obtained via a single trial with some limitations. The delivery of the feasibility trial was feasible; an increase was observed in terms of periodontal health, with a reduction in the gingival inflammation parameters and in the probing pocket depth for patients with gingivitis and periodontitis. In planning future definitive randomized clinical trials, two different placebo-controlled trials should be considered that involve the same separate periodontal disease groups in terms of result interpretation, flexing the participation eligibility criteria, and calculating the statistical analysis based on the number of procedures.

The target time for finding eligible participants was met, but this would not be practical for studies with a large number of participants. We found the eligible participants within one month, but with significant difficulty due to our strict eligibility criteria for our clinic location and patient portfolio. The eligibility of the scanned population was much lower than expected, indicating that the inclusion criteria were strict. Although over-stringent exclusion criteria may increase perceived trial safety, it created difficulty in finding eligible participants in the clinic, leading to cost and time waste. A re-evaluation of the exclusion criteria will increase the applicability of the definitive clinical trial. Smoking criteria are based on evidence that smoking can increase vasoconstriction, which impairs wound healing. However, it was difficult to find an adult patient who never smoked in our clinic's patient portfolio. We estimate that adjusting the smoking criterion to ≤10 per day may increase the rate at which potential patients are found by approximately 20%. Similarly, the condition of not consuming systemic antibiotics within the last 6 months was strict. The reason for this was systemic antibiotics have known properties of increasing host defense and suppressing infection. We believe that if periodontal inflammation is indicative of clinical parameters, the final antibiotic consumption time will be limited to one month, and

the inclusion of these patients in the study will increase the potential participant rate by approximately 40%.

In our trial, all eligible participants who were invited to take part in the study, agreed to participate in the trial. This information is important for planning future studies on oral health. All of the participants decided to enroll when the study protocol was explained.

All of the patients were compliant and attended their appointments regularly. The reason for this is that due to the study's split-mouth design, the whole-mouth SRP took place over two different appointments, and patients were required to attend one control appointment. Of course, as a feasibility study, the fact that the number of participants was kept to a minimum should be taken into consideration. To prevent early withdrawal in studies with a larger number of participants, it will be beneficial to improve the collaboration between researchers on the team and to ensure that doctors can communicate well with their patients.

This trial had several limitations. First, since the researchers had the experience of the first Phase 3 study, they wished to ensure, first, that the methodology allowed the effectiveness of the brushing solutions to be observed as efficiently as possible. To anticipate the needs of a large drug study, the number of participants in this feasibility study was kept to a minimum. Consequently, the probability value was not calculated, and data analyses were presented in percentiles. Second, practical difficulties impeded the histological examination of the participants' periodontal conditions, thus we only focused on macroscopic periodontal healing. Third, microbiological evaluation was not established, as it was considered as a feasibility study. Fourth, some researchers may consider the patients' follow-up time to have been too short. Although it varies according to the severity of inflammation, epithelial healing generally occurs within 7 days, and connective tissue repair with collagen fibers occurs in 21 days. We focused on observing the healing times of two important tissues and therefore preferred to meet with the patients on day 11 (when the brushing solution was finished and the epithelium was repaired) and on day 40 (when the connective tissue had healed after SRP).

## 5. Conclusions

Some promising results were obtained in this feasibility study regarding the improvements in the clinical periodontal parameters of a NitrAdine$^{TM}$-based brushing solution compared to a placebo brushing solution. The use of the split-mouth study design also allowed us to observe both the prophylactic and therapeutic effects of the brushing solution. Long-term randomized clinical trials with a larger sample size supported by microbiological and virological results are needed to confirm the findings regarding the potential success of this material, which can be offered as a proactive option for dentists to differentiate their periodontal treatment strategies.

**Author Contributions:** Conceptualization, B.A. and M.Ö.; methodology, B.A.; software, B.A.; validation, B.A. and M.Ö.; formal analysis, B.A.; investigation, B.A.; resources, M.Ö.; data curation, B.A.; writing—original draft preparation, B.A.; writing—review and editing, B.A. and M.Ö.; visualization, B.A.; supervision, M.Ö.; project administration, M.Ö.; funding acquisition, M.Ö. All authors have read and agreed to the published version of the manuscript.

**Funding:** The materials in this trial and the article processing charge were funded in part by grants from Bonyf AG, Vaduz, Liechtenstein. The trial funders had no role in the trial's design; in the collection, analysis, or interpretation of data, in writing article or in the decision to submit the article for publication.

**Institutional Review Board Statement:** The study protocol was conducted according to the guidelines of the Declaration of Helsinki of 1975, revised in 2013, and approved by both the Istanbul Medipol University Clinical Research Ethics Committee (Number: E-66291034-604.01.01-2798, Date: 15 June 2021) and the Turkish Medicines and Medical Devices Agency (Number: E-68869993-511.06-491724, Date: 28 July 2021).

**Informed Consent Statement:** Written informed consent, including the study protocol and information that the study would be published in an international journal, was obtained from all participants at the beginning of the study.

**Data Availability Statement:** Publicly archived datasets were not used in this study.

**Conflicts of Interest:** The authors declare no conflict of interest.

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
