# Peer review of "Randomized Trial of Feasibility and Preliminary Effectiveness of PerioTabs® on Periodontal Diseases"

_applsci, doi:10.3390/app12031677_

Round 1

Reviewer 1 Report

-Should you present all the results for each patient, the n= 4

-what is the contribution (basic science or clinical applicaction) of the article

Author Response

The file is attached.

Reviewer 2 Report

Thank you very much for letting me review this manuscript, of considerable interest especially on the proactive active ingredient, it needs some revisions
Correct abstract

Correctly entered keywords

Correct materials and methods and statistical analysis

Discussion, limitations must be added, in order to maintain an optimal periodontal state of health for a correct vision of the periodontal ligament it is necessary to motivate the patient to a correct home management through the use of a roto oscillating or sonic toothbrush and toothpastes using hyaluronic acid for keep the periodontium intact and avoid the progression of the disease with the destruction of the tissue itself. I add reference:

DOI 10.3390/ijerph18041468

DOI10.3390/app11188586

To rewrite the conclusions, approaching the various aspects of proactive action

Author Response

The file is attached.

Reviewer 3 Report

Lines 27-34: the paragraph is useless. remove it 

Lines 44- 56: the paragraph is missing of references as well as the fact the oral cavity is not sterile and host a peculiar biofilm which microbial composition changes from the surfaces and the position (tooth biofilm is different from the one in the gingival sulcus, which differs from the one covering implants which is still more different from the one on the tongue) 

https://pubmed.ncbi.nlm.nih.gov/20195365/

https://pubmed.ncbi.nlm.nih.gov/28266111/

https://pubmed.ncbi.nlm.nih.gov/33240361/

https://pubmed.ncbi.nlm.nih.gov/25366221/

https://www.researchgate.net/publication/263888430_Development_of_a_new_protocol_A_macroscopic_study_of_the_tongue_dorsal_surface

Bacteria taxa should be written in italics 

Lines 73-80: paragraph misses references 

sample size is too small 

discussion lacks of discussion with similar studies in literature. 

Author Response

The file is attached.

Round 2

Reviewer 2 Report

The manuscript has been correctly revised, it can be published

Author Response

Dear Reviewer,

Thank you for your detailed comments and suggestions on the major revision. We found them quite useful as we addressed the queries made for the manuscript. We appreciate the time and energy you spent for us.

Reviewer 3 Report

Authors disrespectfully did not agree to the comments. 

I find it very offensive. 

Since they did no comply I reject the manuscript

Author Response

The file is attached.
